# Simulation of Airway Deposition of an Aerosol Drug in COPD Patients

**DOI:** 10.3390/pharmaceutics11040153

**Published:** 2019-04-01

**Authors:** Árpád Farkas, Frantisek Lizal, Jan Jedelsky, Jakub Elcner, Alpár Horváth, Miroslav Jicha

**Affiliations:** 1Centre for Energy Research, Hungarian Academy of Sciences, Konkoly-Thege M. út 29-33, 1121 Budapest, Hungary; 2Energy Institute, Faculty of Mechanical Engineering, Brno University of Technology, Technicka 2896/2, 616 69 Brno, Czech Republic; lizal@fme.vutbr.cz (F.L.); jedelsky@fme.vutbr.cz (J.J.); elcner@fme.vutbr.cz (J.E.); jicha@fme.vutbr.cz (M.J.); 3Chiesi Hungary Ltd., Dunavirág u. 2, 1138 Budapest, Hungary; a.horvath@chiesi.com; 4Department of Pulmonology, County Institute of Pulmonology, 2045 Törökbálint, Hungary

**Keywords:** aerosol drug deposition, inhalation profile measurements, dry powder inhalers

## Abstract

Medical aerosols are key elements of current chronic obstructive pulmonary disease (COPD) therapy. Therapeutic effects are conditioned by the delivery of the right amount of medication to the right place within the airways, that is, to the drug receptors. Deposition of the inhaled drugs is sensitive to the breathing pattern of the patients which is also connected with the patient’s disease severity. The objective of this work was to measure the realistic inhalation profiles of mild, moderate, and severe COPD patients, simulate the deposition patterns of Symbicort^®^ Turbuhaler^®^ dry powder drug and compare them to similar patterns of healthy control subjects. For this purpose, a stochastic airway deposition model has been applied. Our results revealed that the amount of drug depositing within the lungs correlated with the degree of disease severity. While drug deposition fraction in the lungs of mild COPD patients compared with that of healthy subjects (28% versus 31%), lung deposition fraction characteristic of severe COPD patients was lower by a factor of almost two (about 17%). Deposition fraction of moderate COPD patients was in-between (23%). This implies that for the same inhaler dosage severe COPD patients receive a significantly lower lung dose, although, they would need more.

## 1. Introduction

Chronic obstructive pulmonary disease (COPD) is the denomination for a number of lung conditions that cause breathing difficulties. It includes damage to the alveloli (emphysema) and long term inflammation of the small airways (chronic bronchitis). COPD is one of the most frequent airway diseases and is expected to become the third leading cause of death worldwide by 2020 [1]. Medical aerosols emitted by pressurized metered dose inhalers (pMDI) or dry powder inhalers (DPI) are key elements of current COPD therapy. Therapeutic effects are conditioned by the delivery of the right amount of medication to the right place within the airways, that is, to the drug receptors. Inappropriate handling of the inhaler device may cause low pulmonary aerosol drug deposition and low therapeutic effects. The wrong inhalation manoeuvre can also lead to suboptimal drug deposition within the lungs. For instance, too weak inhalation through a DPI device results in a low amount of drug leaving the inhaler and inefficient detachment of drug particles from their large carriers. On the other hand, too forceful inhalation may lead to high throat deposition by impaction and a low dose of the drug deposited in the lungs. Since different inhalers have different internal flow resistances and every patient inhales differently, it is a challenging task to find the best device for each patient, or at least for each category of patients, and to optimize the breathing mode. Computer modeling is one of the promising tools which may help us to predict the deposition of the drug in different regions of the airways [2]. However, for reliable simulation of aerosol drug transport and deposition within the airways, computer models need to be validated against in vitro and in vivo experimental measurements and should use realistic inputs. Application of realistic inputs assumes the knowledge of breathing parameters characterizing individual patients on the one hand, and aerosol parameters characteristic of a given drug on the other hand. It is worth noting that in addition to their high intersubject variability, breathing parameters are also functions of disease severity.

The objective of this work was to measure the realistic inhalation profiles of COPD patients with different degrees of disease severity, simulate the airway deposition distributions of a selected drug (Symbicort^®^ Turbuhaler^®^) with known properties in the airways of the patients, and compare the computed deposition values to similar data of healthy control subjects.

## 2. Materials and Methods

In this work, the estimation of the amount of drug (dose) depositing in different regions of the airways was based on the acquirement of inhalation profiles of different categories of COPD patients while inhaling through Turbuhaler^®^ and determining the aerosol aerodynamic size distributions and tracking of particles with known properties propelled by the airstreams developing in the airways. One of the main reasons why Turbuhaler^®^ inhaler was chosen is that it is one of the most studied devices in the literature. Thus, relevant data needed for numerical modelling is available. The high interest is also due to the fact that Turbuhaler^®^ was the first DPI dispensing doses from a reservoir inside the inhaler [3]. For instance, our literature search revealed that the flow resistance of the device was measured and reported in 30 different works. Some more recent publications include Ciciliani et al., 2017 [4], Buttini et al., 2016, [5] and Azouz et al., 2015 [6]. The mean value of the resistance from these publications is around 65 Pa^0.5^·s·L^−1^, which means that Turbuhaler^®^ is a medium to high resistance inhaler. More detailed technical and design specific information can be found in [7].

### 2.1. Measurement of Breathing Profiles

The breathing manoeuvre of the patients while inhaling the drug through an inhaler is much different from their normal breathing. Patients are advised to exhale without the inhaler, then inhale forcefully through the DPI inhaler, hold their breath for a few seconds (preferably longer than 5 s) and exhale normally, after removing the inhaler. Both the amount of drug and its aerodynamic size distribution are functions of the inhalation flow rate of the patient. Thus, knowledge of the inhalation profile (time-dependent flow rate) is essential in drug delivery modeling.

In this study, inhalation profiles of 47 COPD patients (21 females) were acquired while inhaling through Turbuhaler^®^ DPI inhaler (ethical approval nr. 76-1-20/2017). Since breathing profile is also a function of the patient’s breathing capability, patients were grouped upon disease severity. Based on their previously measured standard spirometric values patients were categorized into four GOLD (Global Initiative for Chronic Obstructive Lung Disease) disease stages [8]. Six patients had mild COPD (stage 1), 20 patients were classified into moderate category (stage 2), 18 patients had severe COPD (stage 3), and 3 patients suffered from very severe COPD (stage 4). Breathing profiles were recorded in a similar way for all categories of patients.

Figure 1 demonstrates the experimental setup. A hand-held spirometer (Otthon Idegen^TM^ mobile spirometer of Thor Laboratories) was placed between the patient’s mouth and the inhaler. The spirometer measured the velocity of the inhaled air at 0.01 s time intervals by the help of inbuilt ultrasound sources and detectors. An important issue is whether the measured velocity value is modified by the spirometer. Indeed, the internal resistance of the spirometer may cause an additional pressure drop, which causes lower air velocities compared to the spirometer-free case. However, spirometers are calibrated to have much lower air resistance than the resistance of the human airways. The used commercial spirometer was validated to comply with this requirement. By the insertion into the system composed of the airways and the spirometer the additional resistance of the inhaler, the resistance of the spirometer becomes even lower compared to the sum of airway and inhaler resistances. The signals of the spirometer were sent to and processed by a computer, and the profile could be interpreted with the help of software delivered together with the spirometer. The registered profiles were then used to construct median profiles for each disease category, except for very severe COPD, which was discounted due to the low statistical power of the results. One median profile was constructed for each disease category. The median profile of each group was obtained by selecting the median value of the flow rates of all patients from the same group at each measurement point (the time resolution was 0.01 s). In addition to the registration of breathing profiles, the breath-hold time after the inhalation was measured for each patient. The exhalation time was not measured because based on our experience its effect on the deposition is very low [9]. Its value was assumed to be 3 s throughout the deposition simulations.

A statistical evaluation of the measured breathing parameters was completed. Standard deviations of the measured quantities were determined. The analysis also included the evaluation of the dependence of peak inspiratory flow through the inhaler (PIF) on patient demographic data, anthropomorphic characteristics, and disease condition. For this purpose, cross-correlation analysis and two-sample t-tests were performed.

### 2.2. Determination of Aerosol Aerodynamic Size Distributions

As mentioned before, the aerosol drug was emitted by the Turbuhaler^®^ inhaler. Several drugs with different aerosol properties are dispensed and commercialized in this inhaler. In this study the characteristics and airway deposition of Symbicort^®^ (AstraZeneca) bicomponent drug (contains two different active pharmaceutical ingredients) was studied, which is a drug commonly used in current COPD therapy of the patients whose FEV_1_ (air volume in the first second of forced expiration) is lower than 70% of the expected normal value and who exacerbate (temporal worsening of patient’s condition) in spite of regular treatment with bronchodilators. Symbicort^®^ contains anti-inflammatory steroid (budesonide, BUD) and bronchodilator (formoterol fumarate dihydrate, FF) components together with lactose monohydrate, which is a carrier.

The two active pharmaceutical ingredients of Symbicort^®^ have different bounding properties. As a consequence, after the deagglomeration of the powder mostly due to particle–particle and particle–inhaler wall collisions during the inspiration, two different aerosol aerodynamic size distributions emerge. Therefore, two different aerosol aerodynamic size distributions characteristic of BUD and FF components need to be determined. In addition, the aerodynamic size distribution of the same drug component depends on the inhalation profile of the patient. Thus, three different aerodynamic size distributions corresponding to the three disease stages exist for each component. Based on the above considerations, six different size distributions were determined in this study. To avoid the difficulties due to different material densities, equivalent aerodynamic particle diameters were considered.

There are several measurement methods for the determination of aerodynamic size distributions of polydisperse particle systems. However, the aerodynamic size distribution of aerosol drugs is regularly measured by cascade impactor techniques in a way prescribed by the Pharmacopoeias. Most of the aerodynamic size distribution measurements of Symbicort^®^ were performed at fixed flow rates ranging from 15 L/min to 90 L/min (e.g., [6,10,11,12]). However, in real life, the inhalation flow rate of the patients is not constant, but it is a function of time. Therefore, the results of the above measurements cannot be directly applied in the present case. Aerodynamic size distributions of Symbicort^®^ corresponding to realistic, measured inhalation profiles were also determined by Bagherisadeghi et al., 2017 [13]. However, the patients were different from the patients in the present study. Thus, adaptation of those size distributions would be questionable. Therefore, a numerical technique for the determination of aerodynamic size distributions that match the inhalation profiles measured in this work was developed consisting in: (i) gathering the results of impactor measurements of different drug size fractions Symbicort^®^ at different constant flow rates; (ii) fitting mathematical functions to the measured data to get the flow rate dependence of each cumulative size fraction; and (iii) determination of cumulative particle size fractions which correspond to the specific inhalation profiles of the present work by the use of these mathematical functions. It is worth noting that in this work all size fractions were expressed as a percent of the dose metered in the device. In this work measured values of cumulative doses of <1 μm, <3 μm, <5 μm, <7 μm, and <10 μm size fractions were plotted as a function of inhalation flow rate (for both BUD and FF components). For this purpose, the citations [6,10,11,12,14,15,16] were used. Examples of flow rate dependent cumulative doses corresponding to <5 μm size fraction derived from the open literature and their linear fits are demonstrated in Figure 2 for BUD and FF drug components. A large particle fraction with an aerodynamic diameter of 15 μm was also defined in this work. This fraction is the dose (mass) fraction represented by particles, which usually deposit in the pre-separator and inlet throat preceding the cascade impactor during the aerodynamic size distribution measurements.

### 2.3. Simulation of Airway Deposition

For the prediction of the amount of drug depositing within different anatomical regions of the airways (upper airways, lungs) the most up to date version of the Stochastic Lung Model (SLM), initially developed by Koblinger and Hofmann, has been applied [17]. The advantage of this model is that it allows the tracking of particles in the whole airway system, which is not possible by using CFD (computational fluid dynamics) techniques. The disadvantage of the SLM model compared to CFD models is that it is based on simple airflows in straight and bent tubes. In addition, the resolution of the whole airway models is lower than those based on CFD techniques in a sense that they can predict the deposition only at regional (e.g., extrathoracic, bronchial, bronchiolar, acinar) level. In the case of SLM model, the maximum resolution is the level of individual airway generations. In this work, deposition in the extrathoracic region was calculated based on the empirical formulas derived by Cheng [18]. Particles which were not filtered out by the upper airways were tracked in stochastic tracheobronchial geometry. Airway lengths, diameters, bifurcation angles, and gravity angles were selected from statistical distributions based on the morphometric database of Raabe et al. [19]. The original diameters were scaled down by 20%, 25% or 30%, depending on the severity of the disease (mild, moderate or severe). The architecture of the acinar airways (where alveoli are present) relied on the data published by Haefeli-Bleuer and Weibel [20]. Morphological changes of the acinar part due to COPD were not considered in this study. Inertial impaction and gravitational settling were considered as deposition mechanisms in both the bronchial and acinar parts of the airways. Since drug particles are usually larger than 1 μm, the contribution of Brownian diffusion to the deposition could be neglected. The SLM model has previously been validated against experimentally measured in vivo aerosol drug deposition results [9].

## 3. Results and Discussion

The results of breathing profile measurements, aerodynamic size distribution computations, and airway deposition simulations are presented in this chapter.

### 3.1. Measured Breathing Parameters and Profiles

The results of the breathing profile measurements described in Section 2.1 are depicted in Figure 3. In the upper left panel, each category of patients is represented by the corresponding median curve (inhalation profile). As the figure demonstrates, both the time of drug inhalation (t_in_) and peak inspiratory flow rate (PIF, the maximum of the curves) decreases with the worsening of disease status. At the same time, the volume of the inhaled air (IV, the area under the curves in Figure 3, upper left panel) during the uptake of medication also decreases with the increase of disease stage number. The corresponding average inhalation flow rate (Q) also becomes lower with the increase of disease severity. Moreover, the measured values of the time interval during which the patients were able to hold their breath after the inhalation (t_b-h_) were lower and lower with the increase of disease severity. Since deposition of aerosols within the airways is highly influenced by the breathing parameters, the evolution of all these breathing parameters as a function of disease severity predicts that patients from different disease stage categories will have different airway deposition fractions and distributions. It is also worth noting that there can be significant differences among the individual breathing curves of the patients from the same group. To highlight the intersubject variability, the standard deviations of the flow rates were also computed at 0.01 s time intervals. The results of this analysis can be seen in Figure 3 (upper right, lower left, and lower right panels).

Average values and standard deviations of the parameters characterizing the breathing profiles demonstrated in Figure 3 (upper left panel) are presented in Table 1, grouped into disease severity categories. A statistical analysis of the measured breathing parameters demonstrated that peak inspiratory flow values were significantly higher for males than for females, but differences upon age, BMI, and disease severity group were not significant (at *p* = 0.05). The peak inspiratory flow measured while the patients inhaled through the device correlated best with the native peak inspiratory flow values measured during normal spirometry. The linear correlation coefficient between the two quantities was 0.51. Intuitively, increased lung resistance may decrease the value of the peak inspiratory flow. However, it was not possible to perform a correlation analysis between patients’ lung resistance PIF, because the used hand-held spirometer did not measure the lung resistance. The decrease in the peak inspiratory flow through the inhaler compared to the native peak inspiratory flow for the same patient is due to the additional flow resistance of the inhaler.

### 3.2. Aersosol Aerodynamic Size Distributions

In addition to breathing parameters, the deposition of medical aerosols is also highly influenced by the inhaled amount and aerodynamic size distribution of the inhaled drug. Since patients from different disease severity categories inhaled with different flow rate, the amount of drug (dose) leaving the device was also different. As Table 2 demonstrates, the emitted dose (ED) was higher for less severe COPD patients, because they were able to perform a more forceful inhalation. The emitted dose is defined as the mass of drug emitted by the inhaler as a percentage of the mass of drug available in the device (metered dose). The rest of the drug remains in the device and its mouthpiece and will not have any therapeutic effect.

The results of aerodynamic size distribution determination are also summarized in Table 2. All mass size fractions are expressed as percentages of the drug mass metered in the device. The values in Table 2 suggest that by the increase of the degree of disease severity not only the amount of inhaled drug is decreasing but also drug particles are becoming larger and larger (lower f_1_, f_3_, and f_5_ values). This is due to the less efficient de-agglomeration of particles inside the inhaler at lower flow rates. The mass fraction of particles with an aerodynamic diameter smaller than 5 μm (f_5_), also called the fine particle fraction, has a special significance in aerosol drug formulation and delivery. It is considered that the higher this indicator, the higher the dose depositing in the lungs is. The values of this parameter predict a lower lung deposition for patients with more severe COPD again. Based on Table 2 the computed mass median aerodynamic diameter (MMAD) values are increasing by the increase of the degree of disease severity. At the same time, the aerodynamic size distributions are becoming more polydisperse (higher geometric standard deviation (GSD) values). This suggests that patients with more severe COPD will receive lower lung doses, but the distribution of the deposited drug will be spatially more uniformly distributed.

### 3.3. Drug Deposition Results

The computed mass of drug depositing in the lungs of COPD patients and healthy subjects as a percent of drug mass metered in the device (lung dose) is presented in Figure 4. The values corresponding to the group of healthy individuals are shown only for comparison and are taken from one of our previous publications [9]. It is worth noting that mild COPD patients (and obviously also the healthy subjects) will not receive Symbicort^®^ drug, their results are shown here only for comparison purpose.

As expected, severe COPD patients performing the lowest flow rate and inhaling the lowest amount of drug will have a significantly lower deposited lung dose. While the computed dose of the drug in the lungs of mild COPD patients compares with that of healthy subjects, the lung dose characteristic of severe COPD patients is lower by a factor of almost two. It is expected that this tendency holds for very severe COPD patients, as well. The whole distribution of the drug in the device (DEV), extrathoracic airways (ET), lungs (LUNG), and the exhaled fraction (EXH) are summarized in Table 3. As the table demonstrates, the dose fraction depositing in the device decreases as the degree of disease severity decreases. Less severe COPD patients have less impaired lung function, and they can inhale more forcefully while taking the drug. Therefore, more drug leaves the device and less drug remains in it. The extrathoracic dose seems to be less sensitive to the disease severity, though some increase with the decrease of disease severity can be observed. Actually, this dose fraction is a result of two competing effects. On the one hand, less severe COPD patients have higher emitted doses and higher deposition in the throat due to higher air and particle velocities. On the other hand, the emitted particles are smaller and have higher chances to penetrate into the lungs. The exhaled fraction is lower for less diseased patients due to the higher extrathoracic and lung deposition.

## 4. Conclusions

This study demonstrated that computer modelling based on realistic input data can provide valuable information on the deposition distribution of aerosol drugs within the airways of COPD patients with different degrees of disease severity. Based on the current results, for the same inhaler dosage, severe COPD patients receive a significantly lower lung dose than moderate COPD patients, therefore, they would need more.

## Figures and Tables

**Figure 1 pharmaceutics-11-00153-f001:**
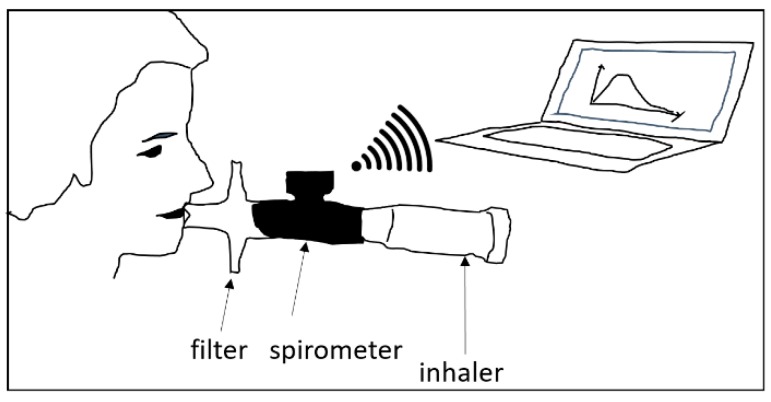
Scheme of inhalation profile measurements.

**Figure 2 pharmaceutics-11-00153-f002:**
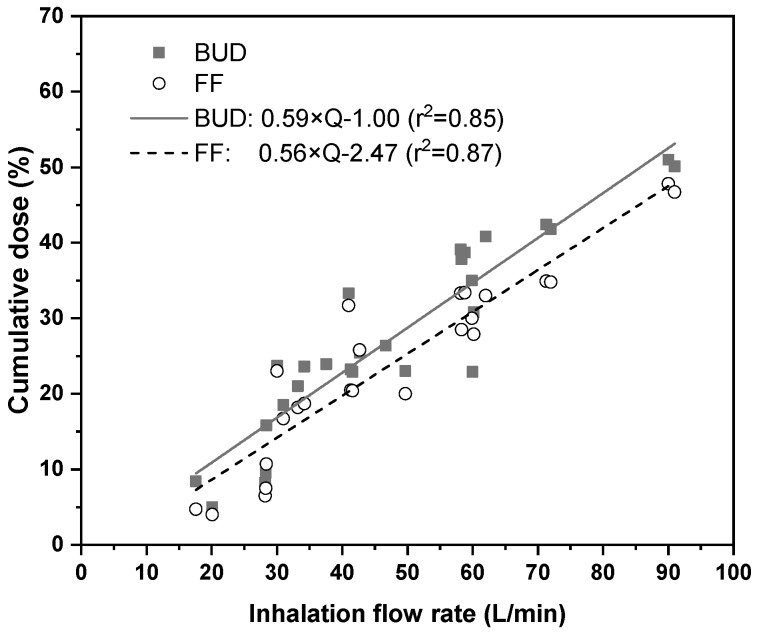
Cumulative dose values (expressed as percentage of the metered dose) of BUD (budesonide) and FF (formoterol) represented by drug particles with aerodynamic diameters smaller than 5 μm gathered from the literature ([6,10,11,12,14,15,16]) and linear functions fit to the corresponding datapoints. Q—inhalation flow rate.

**Figure 3 pharmaceutics-11-00153-f003:**
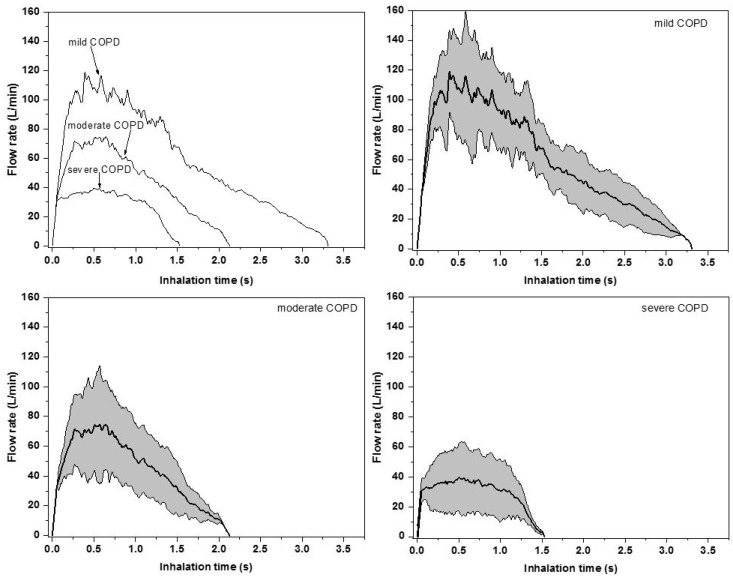
Median inhalation curves of mild, moderate, and severe chronic obstructive pulmonary disease (COPD) patients while inhaling through the inhaler (upper left panel) and their standard deviations (upper right, lower left, and lower right panels).

**Figure 4 pharmaceutics-11-00153-f004:**
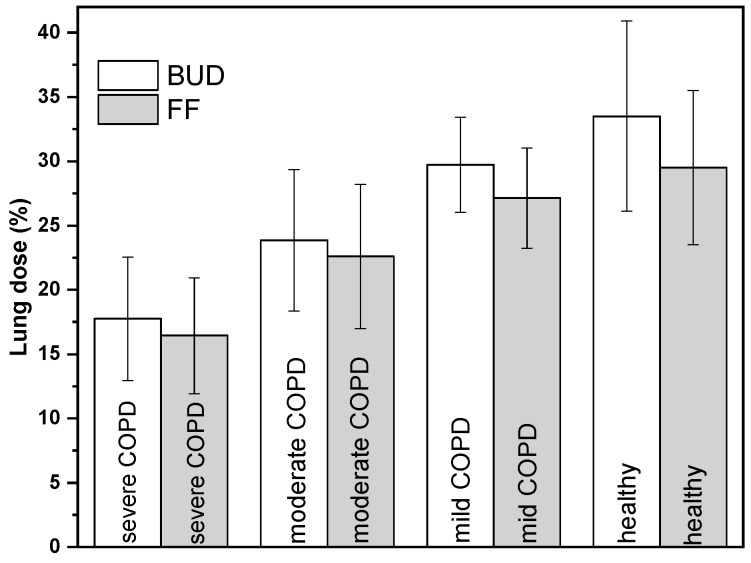
Lung doses of severe, moderate, and mild COPD patients in comparison with lung dose of healthy subjects. The lung dose is expressed as a percent of the dose metered in the device. BUD—budesonide; FF—formoterol fumarate dihydrate.

**Table 1 pharmaceutics-11-00153-t001:** Average breathing parameters of chronic obstructive pulmonary disease (COPD) patient groups while inhaling through Turbuhaler^®^. PIF—peak inspiratory flow rate, IV—inhaled volume; tin—inhalation time; t_b-h_—breath-hold time; Q–mean flow rate during the inhalation.

Breathing Parameter	Severe COPD	Moderate COPD	Mild COPD
**PIF (L/min)**	39 ± 22.6	75 ± 40.2	119 ± 30.9
**IV (L)**	0.7 ± 0.2	1.6 ± 0.8	3.3 ± 0.6
**t_in_ (s)**	1.5 ± 0.5	2.1 ± 1.1	3.3 ± 0.5
**t_b-h_ (s)**	6.4 ± 4.2	7.2 ± 3.5	8.8 ± 2.9
**Q (L/min)**	28 ± 9.5	45.7 ± 18.1	60 ± 16.3

**Table 2 pharmaceutics-11-00153-t002:** Computed budesonide (BUD) and formoterol (FF) particle size fractions characteristic of different disease severity groups. ED—emitted dose; f_1_, f_3_, f_5_, f_7_, and f_10_—mass size fractions of <1 μm, <3 μm, <5 μm, <7 μm, and <10 μm particles, respectively; f_large_—fraction of large particles; MMAD—mass median aerodynamic diameter; GSD—geometric standard deviation.

Aerosol Parameter	Severe COPD		Moderate COPD		Mild COPD	
	BUD	FF	BUD	FF	BUD	FF
ED (%)	49.74	37.13	60	51.29	68.30	62.73
f_1_ (%)	1.87	1.83	3.64	3.25	5.07	4.39
f_3_ (%)	8.33	6.09	18.95	16.53	27.53	24.97
f_5_ (%)	16.46	15.77	27.56	25.56	36.53	33.47
f_7_ (%)	21.73	22.55	34.11	32.93	42.44	38.82
f_10_ (%)	23.05	25.23	36.09	35.68	44.85	40.49
f_large_ (%)	26.69	11.9	23.91	15.61	23.45	22.24
MMAD (μm)	3.03	3.44	2.67	2.87	2.45	2.43
GSD (–)	2.02	2.23	1.87	1.91	1.81	1.80

**Table 3 pharmaceutics-11-00153-t003:** Computed deposition fractions with standard deviations in the inhaler (DEV), upper airways (ET), lungs (LUNG, bronchial, bronchiolar and acinar airways) and the exhaled fraction (EXH) of both drug components in different disease category groups.

Dose Fractions	Severe COPD		Moderate COPD		Mild COPD	
	BUD	FF	BUD	FF	BUD	FF
DEV (%)	50.3 ± 8.1	62.9 ± 7.3	40.0 ± 4.3	48.7 ± 4.4	31.7 ± 5.1	37.3 ± 4.8
ET (%)	27.3 ± 4.6	16.3 ± 4.5	31.7 ± 3.8	24.7 ± 3.7	34.8 ± 4.8	32.2 ± 5.0
LUNG (%)	17.7 ± 4.8	16.4 ± 4.5	23.8 ± 5.5	22.6 ± 5.6	29.7 ± 3.7	27.1 ± 3.9
EXH (%)	4.6 ± 1.1	4.4 ± 1.8	4.5 ± 1.6	4.1 ± 1.9	3.8 ± 1.6	3.4 ± 1.3

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
