# Peer review of "Simulation of Airway Deposition of an Aerosol Drug in COPD Patients"

_pharmaceutics, 2019, doi:10.3390/pharmaceutics11040153_

Reviewer 1 Report

The field of the work is very interesting and promising in the pharma sciences. From my point of view, the article is well written and comprehensible. The main critical remark I have is related to the statistical analysis that is missing. 

Please find below some remarks and suggestions:

1.     Line 43. I suggest “….low lung dose of the drug deposited in the lungs.”

2.     Sentence 89-91. Please add a reference.

3.     The spirometer is placed between the patient’mouth and the device. Is the velocity that should be obtained without spirometer (i.e. commom use of the device) are similar as in the presence of it. In other word, does the spirometer affect the measurement by itself? Could you comment that?

4.     More info on the device should be given (e.g. resistance, reservoir) or referenced (e.g. design)

5.     The terms ‘size’ and ‘aerosol size’ are used interchangeably throughout the manuscript (e.g. paragraph 105-116). This is confusing. I suggest to use the term ‘aerodynamic size’, which is commonly accepted in the field. Please modify throughout the manuscript.

6.     Line 118. Please revise the units throughout the manuscript.

7.      The method for measuring size distributions should be more described (how to obtain the mentioned fraction? The flow rate? etc.). Why were conventional approved assays (cascade impactors), described in the pharmacopeia, not use in this work?

8.     More comments should be added regarding the comparison of (i) particle size fractions characteristic of different disease severity groups and (ii) lung doses.

9.     From my point of view, appropriate statistical analysis should be performed for publication.

Author Response

See the attachment.

Sincerely,

Arpad Farkas

Reviewer 2 Report

The manuscript reports different breathing profiles of COPD patients with varying (mild, moderate and severe) disease severity, when using a Turbuhaler inhaler. Authors then used these breathing conditions together with extrapolated aerosol diameter distributions for two active drug compounds in a   computational lung deposition model  to estimate the deposition of these compounds inside patient lungs.

The measured patient breathing profiles are helpful for lung deposition modeling research.The following are some of the areas where I would have appreciated additional clarifications.

 Considering personalized medicine, it will be helpful if the authors can report the statistical variation in the breathing wave-forms, may be in terms of standard deviation. 

Will it be possible to correlate the inhalation waveforms with the Turbuhaler resistance and patient lung resistance or any spirometic parametrers)?

In COPD patients many of the airways will be obstructed. Apart from the reduced inhalation flow-rates and associated particle diameter distribution, can the stochastic lung deposition model be modified to represent the obstructed airways?

Tabl2 3: The DEV(%) deposited in the inhaler is same as the emitted dose ED(%) in Table 2 ?

It is not mentioned whether the reported % is in terms of the emitted dose?

Fig 2: Would be better if it is mentioned that the Lung dose (%) is thoracic+extrathoracic deposition.

Minor comments:

Missing micron in different locations.

Author Response

See the attachment.

Sincerely,

Arpad Farkas

Round  2

Reviewer 1 Report

Thank you for considering my remarks and suggestions. This parer is of great interest in the Pharma field nowadays. Congratulations for this work.